# Perinatal Outcomes at Birth in Women Infected and Non-Infected with SARS-CoV-2: A Retrospective Study

**DOI:** 10.3390/healthcare11212833

**Published:** 2023-10-27

**Authors:** Rafael Vila-Candel, Anna Martin-Arribas, Enrique Castro-Sánchez, Ramón Escuriet, Jose M. Martin-Moreno

**Affiliations:** 1Sciences Faculty, Universidad Internacional de Valencia—VIU, 46002 Valencia, Spain; rafael.vila@professor.universidadviu.com; 2Department of Obstetrics and Gynecology, Hospital Universitario de la Ribera, 46600 Alzira, Spain; 3Foundation for the Promotion of Health and Biomedical Research in the Valencian Region (FISABIO), 46020 Valencia, Spain; 4Ghenders Research Group, School of Health Sciences Blanquerna, Universitat Ramon Lull, 08025 Barcelona, Spain; rescuriet@gencat.cat; 5College of Business, Arts, and Social Sciences, Brunel University London, Uxbridge UB8 3PH, UK; enrique.castro-sanchez@brunel.ac.uk; 6Health Protection Research Unit, Healthcare-Associated Infections and Antimicrobial Resistance, Imperial College London, London SW7 2BX, UK; 7Research Group on Global Health and Human Development, University of the Balearic Islands, 07122 Palma de Mallorca, Spain; 8Catalan Health Service, Government of Barcelona, 08014 Barcelona, Spain; 9Department of Preventive Medicine and Public Health, Universitat de València, 46010 Valencia, Spain; 10Biomedical Research Institute INCLIVA, Clinic University Hospital, 46010 Valencia, Spain

**Keywords:** SARS-CoV-2, vaccination, women’s health, perinatal outcomes, adverse outcomes

## Abstract

Background: Coronavirus disease 2019 (COVID-19) was declared as a pandemic and public health emergency on 11 March 2020 by the World Health Organization. Different clinical trials on the efficacy of mRNA vaccination have excluded pregnant women, leading to a lack of empirical evidence on the efficacy of the vaccine in this population. The aim of the study was to examine the association between severe acute respiratory syndrome coronavirus 2 (SARS-CoV-2) infection at birth and adverse perinatal outcomes in infected and non-infected women from a university hospital in Spain. Methods: The data were obtained from electronic health records from 1 March 2020 to 28 February 2022. A bivariate descriptive analysis was performed, comparing women with and without confirmed SARS-CoV-2 infection during pregnancy using the chi-square test. A multivariate logistic regression was complementarily conducted to determine whether SARS-CoV-2 infection increases the risk of adverse obstetric and perinatal outcomes. Results: A total of 2676 women were divided into two groups: non-infected with SARS-CoV-2 (*n* = 2624) and infected with SARS-CoV-2 (*n* = 52). Infected women were primarily multiparous (*p* < 0.03) and had received an incomplete vaccination regimen (*p* < 0.001). A greater incidence of premature rupture of membranes (*p* < 0.04) was observed among the non-infected women. Pertaining to perinatal outcomes, there was a notable rise in NICU admissions (*p* < 0.014), coupled with an extended duration of stay (*p* < 0.04), for neonates born to infected mothers in comparison to their non-infected counterparts. Conclusion: Although SARS-CoV-2 infection may pose significant risks to pregnant women and their infants, adverse obstetrical/puerperal outcomes do not significantly differ between women infected and non-infected to SARS-CoV-2 in our study. NICU admissions were higher for neonates born to infected mothers. Additionally, coronavirus disease 2019 vaccination during pregnancy is not associated with severe adverse perinatal outcomes.

## 1. Introduction

Coronavirus disease 2019 (COVID-19) was declared as a pandemic and public health emergency on 11 March 2020 by the World Health Organization. Since the first detected case of COVID-19 in Wuhan, Hubei, China, in December 2019, the infection spread to the rest of the world with an alarming number of cases [1,2].

The initial transmission pattern has been suggested to be zoonotic, while the current spread has been from person to person through airborne transmission following close contact with an infected person or direct contact with contaminated surfaces [3,4]. The risk of vertical transmission appears to be low (approximately 0–4%) and, therefore, of modest relevance [5,6,7]. Severe acute respiratory syndrome coronavirus 2 (SARS-CoV-2) may be detected in amniotic fluid, but this is exceptional. Although SARS-CoV-2 has been isolated in the placenta, vertical transmission of the virus appears to be rare and limited to cases of severe maternal infection [8]. Most described cases of infection in newborns are from horizontal transmission. In addition, the virus has not been detected in vaginal secretions or breast milk [8,9,10]. The available data suggest a range of viral RNA presence in milk samples, spanning from 2% to 6%. A recent systematic review [11] of lactating individuals affected by COVID-19 revealed a 13.2% detection rate of SARS-CoV-2 RNA. Similarly, another systematic review [12] indicated around a 2% detection rate of RNA in breast milk. Among these studies, the most extensive involved 110 women from the US [13], 65 of whom tested positive for SARS-CoV-2. In this study, 6% of the milk samples exhibited SARS-CoV-2 RNA presence; however, no infectious particles were cultured from these samples. Notably, a recent clinical trial yielded different results, as no SARS-CoV-2 RNA was detected in any of the breast milk samples [14]. The findings of this trial endorse official recommendations that underline the safety of breastfeeding during COVID-19. This perspective prioritizes breastfeeding due to its potential to confer maternal–neonatal benefits.

COVID-19 may be asymptomatic in up to 75% of pregnant women [15]. When symptoms appear, the infection is classified according to the severity of respiratory symptoms as mild, moderate, and severe [4]. The majority of symptomatic cases during gestation present a mild infection (85%) [16]. The most frequent symptoms during pregnancy are fever (40%) and cough (39%), while other less frequent symptoms are myalgia, dyspnea, odynophagia, anosmia, expectoration, headache, and diarrhea [5].

Pregnant women are at a higher risk of severe infection-related complications with respect to the non-pregnant population, especially in the third trimester and when the following risk factors are present: advanced maternal age, high body mass index (BMI of >30 kg/m^2^), chronic hypertension, and/or pregestational diabetes [17]. Approximately 15% of COVID-19 cases progress to severe forms [18]. Approximately 5% of infected pregnant women may require admission to an intensive care unit (ICU), and 3% may require invasive ventilation [7]. The rate of mortality in pregnant women ranges from 0.1% to 1.2%. Severe forms present the following main complications: severe pneumonia, acute respiratory distress syndrome, thromboembolic disease, bacterial respiratory superinfection, cardiac alterations, encephalitis, sepsis, and septic shock [6,7,19].

Regarding fetal or neonatal complications, the current data do not suggest an increased risk of miscarriage or early gestational loss in pregnant women with COVID-19 [7]. Similarly, severe acute respiratory syndrome coronavirus and Middle East respiratory syndrome coronavirus have not been reported to demonstrate a clear causal relationship with these complications [20]. No increased risk of congenital defects has also been described [9]. The main perinatal complication associated with COVID-19 is prematurity, with rates around 20%, mainly at the expense of iatrogenic prematurity [5,7]. There may be placental involvement and anatomopathological alterations in the form of vascular malperfusion or intervillous fibrin deposits whose consequences at the fetal level have yet to be determined [5]. No significant differences in other perinatal outcomes have been found among neonates born to mothers with COVID-19, although 25% are admitted to a neonatal care unit [8]. Regarding neonatal COVID-19, 50% of cases generally present to adult clinics with comparable results in terms of symptomatology and analytical and imaging findings [7].

Between 2020 and 2022, Spain experienced six pandemic waves of COVID-19, each presenting distinct challenges. Throughout this period, stringent measures were enforced during states of emergency to curb viral transmission [21]. These measures had direct repercussions on the medical care provided to pregnant women, resulting in discernible adjustments to obstetric protocols [22,23]. Scientific societies remained dynamic, continually adapting obstetric care protocols in response to evolving epidemiological trends [5]. The intricate interplay between COVID-19 prevention and ensuring safe care for mothers and newborns assumed paramount importance.

However, across these successive pandemic waves, certain indispensable practices in maternal and childbirth care underwent necessary adaptation or suspension to mitigate virus spread. As the landscape evolved, practices like labor companionship, early skin-to-skin contact, and rooming-in were supplanted by measures such as mother–infant separation and neonatal intensive care unit (NICU) admission [23]. While these adjustments aimed to safeguard both mothers and medical personnel, concerns about care quality inevitably arose.

The modification of these practices bore negative implications for the maternal experience and neonatal health, hampering mother–infant interaction and impeding breastfeeding promotion [24]. As the pandemic progressed, healthcare professionals persevered in their efforts to strike a delicate balance between preventive measures and maintaining a warm, secure environment during the birthing process. This underscored the urgency of embracing adaptable, evidence-based approaches to safeguard maternal and neonatal health amid health crises and epidemiological fluctuations [22].

In 2021, the vaccination campaign first expanded to encompass adults, and subsequently, commencing December of that year, extended to include pregnant women [25]. Robust studies unequivocally endorsed the safety and efficacy of vaccines, driving a gradual acceptance and favorable response within the populace [6,9,26]. This multifaceted interplay between pandemic waves, evolving protocols, and vaccination efforts underscores the need for dynamic, evidence-driven strategies to ensure optimal care provision during times of crisis.

Although a published series of vaccinations during pregnancy still include few cases, the currently available COVID-19 vaccines are not expected to pose a problem during pregnancy and lactation [6,9,26,27]. The recent recommendation is to offer a messenger RNA (mRNA) vaccine to all pregnant women following established vaccination schedules and especially to pregnant women with comorbidities (e.g., patients who underwent transplantation, who are immunosuppressed, or who have cardiopulmonary, renal, oncologic, or other conditions) [28]. The ideal time for administration is the second trimester. However, if the epidemiological risk is high or there are comorbidities, there is no inconvenience in administering the vaccine in the first trimester [29].

Different clinical trials on the efficacy of mRNA vaccination have excluded pregnant women, leading to a lack of empirical evidence on the efficacy of the vaccine in this population [26,30]. Therefore, vaccine safety and efficacy during pregnancy are mainly evaluated through observational epidemiological studies [31].

In this study, at the beginning of the pandemic, where vaccination efforts were nascent and the circulating virus reached considerably higher levels within the community, our aim was to explore the association between SARS-CoV-2 infection at birth and adverse obstetrical–neonatal outcomes from a university hospital in Spain.

In addition, we assessed the proportion of neonates with reverse transcription polymerase chain reaction (RT-PCR)-detectable SARS-CoV-2 from all births among women with COVID-19 diagnosed at the onset of labor.

## 2. Materials and Methods

This observational, retrospective study (clinical/epidemiological, descriptive, and analytical in nature) was conducted among pregnant women who visited the Hospital Universitario de la Ribera (HULR) for delivery from 1 March 2020 to 28 February 2022.

The HULR is a regional hospital with a population area of 250,000 inhabitants and assists an average of 1300 deliveries per year. Deliveries of less than 34 weeks of gestation are referred to a referral hospital.

The women were allocated into infected and non-infected groups based on the results of RT-PCR or antigen testing (nasopharyngeal exudate) for SARS-CoV-2 at hospital admission. The infected group included women positive for SARS-CoV-2 on RT-PCR or antigen testing during delivery, and non-infected group included women with a negative result.

Information on COVID-19 vaccination was collected from both unvaccinated and vaccinated women. Women who received at least one dose of the COVID-19 vaccine either before or during the current pregnancy were included in the vaccinated cohort. Although this group was labeled as “vaccinated”, some women did not follow the full vaccination schedule at the time of delivery.

Pregnant women infected with SARS-CoV-2 admitted for medical/surgical reasons other than childbirth were excluded.

### 2.1. Sample Size

From 1 March 2020 to 28 February 2022, the entire population whose deliveries were attended during the study period was considered. In the first year (1 March 2020, to 28 February 2021), there were 1236 deliveries. Similarly, in the second year (1 March 2021, to 28 February 2022), there were 1440 deliveries.

### 2.2. Data Collection

The information necessary for inclusion in the study was obtained from two different sources: (1) electronic medical records of specialized care, from which variables related to care during childbirth and puerperium and subsequent complications were collected, and (2) primary care medical records, from which the vaccination status.

The obstetrical outcomes evaluated were as follows: premature rupture of membranes (PROM), preterm birth (<37 weeks of gestation), placental abruption, antepartum hemorrhage, postpartum hemorrhage, cesarean section, instrumental delivery, fetal distress (defined by a healthcare provider), fetal growth restriction (estimated fetal weight below the third percentile), pregnancy-induced hypertension, and gestational diabetes.

The COVID-19-related outcomes were presenting signs/symptoms, admission to an intensive care unit (ICU), length of ICU stay of more than 4 days, intubation, supplemental oxygen, cardiac manifestations (myocardial infarction, cardiomyopathy, or arrhythmia), neurologic manifestations (seizures, hemorrhagic or ischemic stroke, or coma), thrombotic manifestations (deep vein thrombosis, pulmonary embolism, or arterial thrombosis), coagulopathy, and maternal death during admission.

The neonatal outcomes were small for gestational age (SGE: birthweight below the 10th percentile), large for gestational age (LGE: birthweight above the 90th percentile), Apgar score of <7 at 5 min, neonatal intensive care unit (NICU) admission, length of neonatal ICU stay of more than 4 days, respiratory distress, ventilator support, SARS-CoV-2 infection, hypoxic ischemic encephalopathy, and neonatal death.

Other covariates included sociodemographic and medical characteristics that could act as potential risk factors: maternal age, parity, gestational age at birth, date of birth, country of origin, gestational pathologies (thyroid, gestational diabetes and preeclampsia), other clinical variables (maternal obesity, asthma, or smoker), mRNA SARS-CoV-2 vaccine (Pfizer-BioNTech), number of doses, and time between last vaccine administration and infection (when infected) (Figure 1).

### 2.3. Statistical Analysis

We first performed a univariate descriptive analysis of the birth characteristics. Quantitative variables were summarized as means and standard deviations and categorical variables as absolute and relative frequencies.

Subsequently, a bivariate descriptive analysis was conducted, comparing the infected and non-infected groups. Categorical variables were compared using the Chi-square test and quantitative variables using ANOVA. The computation of odds ratios and their corresponding confidence intervals, accompanied by *p*-values derived from log-odds through the utilization of the Wald test, was executed by employing a series of logistic regression models featuring a sole independent variable [32]. In the investigation pertaining to distinct vaccination statuses (unvaccinated, incomplete regimen, complete regimen), an analogous methodology was applied. However, in this instance, a multinomial regression framework with a logit link function was employed. Finally, a multivariate logistic regression analysis using the backward Wald method was performed to determine whether SARS-CoV-2 infection increases the risk of adverse obstetric and perinatal outcomes. The significance of SARS-CoV-2 infection for each outcome variable and ORs were evaluated. Statistical significance was set at a *p*-value of ≤0.05. Data were statistically analyzed using R (version 4.0.2).

### 2.4. Ethics Statement

Neither informed consent nor a patient information sheet was required owing to the retrospective nature of the study. Only electronic medical records were reviewed, and no contact was made at any time with the patients whose data were analyzed. No identifying data of the women and/or their newborns were included in the data collection notebook. The study complied with the Helsinki Recommendations for biomedical studies and was approved by the Research and Ethics Committee of the HULR (HULR_2022_56).

## 3. Results

### 3.1. Descriptive Analysis

We obtained a total sample of 2676 women and divided them into two groups: infected with SARS-CoV-2 (*n* = 52) and uninfected with SARS-CoV-2 (*n* = 2624).

The mean maternal age was 31.2 ± 6.1 years; of the women, 55.5% were nulliparous (*n* = 1484), and 71.8% were born in Spain (*n* = 1922). SARS-CoV-2 positivity was determined via RT-PCR (*n* = 50) and antigen testing (*n* = 2). The incidence of SARS-CoV-2 infection was 14.6 per 1000 births (18/1236) in 2020 and 23.6 per 1000 births (34/1440) in 2021. A total of 20.5% of the sample was vaccinated (*n* = 550): 16.5% with a complete Pfizer regimen (*n* = 442) and 4.0% with at least one dose (*n* = 108). The median number of days from the last vaccine administration to infection was 102.5 days with an interquartile range of 72.75 days.

From the beginning of 2021, SARS-CoV-2 surveillance in Spain included genomic information assessment for confirmation of the presence of variants using sequencing techniques. The information available in the surveillance system in Spain (SiViEs) is analyzed on a weekly basis. This study collected all cases recorded during the first five waves in Spain. According to the SiViEs data, the first three waves (from March 2020 to January 2021) were caused by the alpha variant, the fourth wave (from July to September 2021) by the delta variant, and the fifth wave (from January 2022) by the omicron variant.

Thus, the distribution of cases according to the estimation of SARS-CoV-2 variants was as follows: alpha, 38.5% (20/52); delta, 7.7% (4/52); and omicron, 53.8% (28/52).

### 3.2. Bivariate Analysis

Table 1 shows the results of the bivariate analysis between the infected and non-infected groups. In total, 1.9% of the sample (*n* = 52) was positive for SARS-CoV-2. We observed that the risk of infection of unvaccinated women was four times lower compared to vaccinated women (OR: 4.0 (95% CI: 2.3–6.9; *p* < 0.001). In contrast, when grouping the sample in women vaccinated with a complete regimen (two doses) and incomplete regimen (one dose), the risk estimate suggests that women with incomplete vaccination had five-fold risk compared with those vaccinated with the complete dose (OR: 5.6, 95% CI: 2.2–13.0; *p* < 0.001). Multiparous women were significantly more infected than nulliparous women (OR: 1.9 95% CI: 1.1–3.3; *p* = 0.03).

The sociodemographic variables did not significantly differ between the groups.

Table 2 shows the comparison of the obstetric variables and their exposure to SARS-CoV-2. The data indicate that infected women had a lower risk of developing PROM compared with non-infected women (OR: 0.4 95% CI: 0.2–0.7; *p* = 0.004). Only 7.7% (4/52) of the women positive for SARS-CoV-2 presented minimal symptoms and, therefore, no antepartum or postpartum complications. The cesarean section rate was 19.7% (527/2676) among the total population, 15.4% (8/52) among the infected group, and 19.8% (519/2624) among the non-infected group; no significant differences were found between the groups. No cases were observed of fetal growth restriction, placental abruption, maternal death, maternal intubation, and cardiac, neurological, thrombotic, and coagulopathy manifestations. We observed that infected women were admitted to ICU for more days (*p* < 0.001).

On the other hand, we analyzed the differences among various maternal comorbidities, such as maternal obesity (BMI > 30 kg/m^2^), asthma, hyperthyroidism, hypothyroidism, and smoking, and did not observe statistically significant differences between infected and non-infected women.

The outcomes and complications in the newborns born to mothers with COVID-19 did not also significantly differ between the groups (Table 3). No cases of neonatal death, neonatal anemia, neonatal sepsis, neonatal SARS-CoV-2 infection, Apgar < 7 at 5 min, and hypoxic ischemic encephalopathy were observed in infected women. Newborns born to infected mothers did have a higher likelihood of NICU admission compared to those born to uninfected mothers (*p* = 0.014) and a length of stay at NICU more than four days (*p* = 0.004).

We also analyzed sociodemographic, obstetric, and neonatal outcomes among vaccinated and unvaccinated women (Appendix A). Multiparous women and women born outside Spain were vaccinated less than women born in Spain (*p* < 0.001; *p* = 0.016, respectively), with no statistically significant differences in terms of maternal age. Regarding obstetric–neonatal outcomes, the rate of preterm birth (<37 weeks), preeclampsia, oxygen supplementation in women during ICU admission, and presenting signs and symptoms of COVID-19 were significantly higher in unvaccinated women compared to vaccinated women *(p* = 0.035; *p* < 0.001; *p* = 0.002; and *p* = 0.007, respectively).

A subgroup analysis of the infected group (*n* = 52) was then performed to analyze the differences between the vaccinated and unvaccinated women. In the analysis among the women infected with SARS-CoV-2 (Appendix A), we found no significant differences between the socio-demographic characteristics and vaccination status. Although vaccinated women were slightly older than unvaccinated women (32 ± 6.8 vs. 30 ± 6.3) and with origin of Spain (73.1 vs. 65.4), the differences were not statistically significant (*p* = 0.271; and *p* = 0.548, respectively). There were no cases of placental abruption, fetal growth restriction, maternal sepsis, maternal death, intubation, COVID-19 complications, newborns with Apgar < 7 at 5 min, neonatal sepsis, neonatal infection with SARS-CoV-2, hypoxic ischemic encephalopathy, and neonatal death in any of the groups.

Although the differences found were not statistically significant, a higher number of preterm deliveries (<37 weeks), PROM, cesarean sections, maternal and neonatal admission to the ICU/NICU was observed in vaccinated women and their newborns compared to unvaccinated women. On the other hand, a relationship was observed between gestational diabetes and being vaccinated (*p* = 0.037).

Furthermore, we were interested in examining the disparities among vaccination statuses (unvaccinated, incomplete regimen, and complete regimen) and the variables considered in the study (Appendix A). Certain significant findings were observed: older women were more likely to have completed their vaccination regimen (*p* < 0.001), primiparous women had a higher likelihood of being vaccinated in complete regimen (*p* < 0.001), and foreign women exhibited a lower propensity for being vaccinated (*p* < 0.001). Women with a complete vaccination regimen displayed a significantly heightened risk of experiencing postpartum bleeding, with a five-fold increase compared to the unvaccinated (*p* < 0.001). Additionally, a two-fold increase in the risk of hypertensive disorders during pregnancy was identified among women with a complete vaccination regimen (*p* = 0.006).

### 3.3. Logistic Regression Analysis

The multivariate logistic regression analysis aimed to determine whether SARS-CoV-2 infection and vaccination status increases obstetric and perinatal risks. This entailed evaluating the statistical significance of independent variables as potential predictors of various outcomes, including preterm birth (<37 weeks’ gestation), premature rupture of membranes (PROM), cesarean section (C-section) delivery, as well as admission to both the intensive care unit (ICU) and neonatal intensive care unit (NICU). *p*-values of <0.05 indicated whether COVID-19 was a risk/protective factor for each variable. Table 4 shows the results of the regression analysis, including the calculated ORs.

We observed that the presence of SARS-CoV-2 infection demonstrated a noteworthy connection to reduced chances of PROM (*p* = 0.003), while simultaneously showing an elevated likelihood of NICU admission (*p* = 0.009). Vaccination status revealed a statistically significant increase in PROM odds (OR = 1.2; 95% CI 1.0–1.47; *p* = 0.048). On the other hand, we were interested in exploring the relationship between prematurity and PROM, observing a direct association (*p* = 0.0015), as well as with maternal admission to the ICU (*p* = 0.003) and neonatal admission to the NICU (*p* < 0.001). Infected mothers did not exhibit a higher likelihood of ICU admission compared to non-infected ones, but the risk increased by five-fold (OR 5.6, 95% CI: 1.6–15.5) when the delivery was preterm. Similarly, infants born to mothers with preterm deliveries faced a six-fold risk (OR 6.7, 95% CI: 4.7–9.4) of NICU admission compared to those born at full-term.

## 4. Discussion

This study adds to the body of evidence that our results showed no significant differences in perinatal outcomes between SARS-CoV-2-positive and -negative women, and vaccination against COVID-19 during pregnancy is safe and does not affect either obstetric or neonatal outcomes. In our sample, the risk of postpartum complications and ICU admission in the women infected with SARS-CoV-2 was low, and the maternal and neonatal mortality rates did not increase. Vaccinated and unvaccinated women were equally exposed to the indirect effects of the pandemic during both pregnancy and delivery.

### 4.1. Obstetric and Neonatal Outcomes between Infected and Non-Infected Women

The prematurity rate observed in our center was 7.6%, close to the pre-pandemic rate (6.4–7%) [23,33]. We did not find a significant relationship between the outcomes and infection, possibly owing to the small number of cases managed in our hospital; pregnant women with threatened preterm deliveries were referred to the referral hospital, and only deliveries of more than 34 weeks of gestation or preterm babies whose transfer was impossible owing to obstetric conditions of an imminent delivery were managed in our center. The observed rate was lower than that reported by other authors who relate the increase in the figures close to 20% to iatrogenic management of deliveries of mothers with COVID-19 [5,7].

The total cesarean section rate was low in the study population (19.6%), but not increased in the infected group (15.4%), in contrast with the results of some systematic reviews (around 30%) [17] or other recent reports (55%) [7]. This increase in the usage of cesareans could be related to the potential increased maternal and fetal concerns and/or the exposure risk to healthcare professionals [19,34,35]. The observed lower rate of cesarean sections among infected women implies that the infection may have presented with mild symptoms, taking into account the heterogeneity in therapeutic management of COVID-19 at that time among neighboring countries [21].

Between the infected and non-infected pregnant women in our study, no significant differences were obtained in terms of serious complications, neither at admission nor during puerperium (abruptio placentae, antepartum/postpartum hemorrhage, fetal distress, fetal growth restriction, preeclampsia, gestational diabetes, or maternal death) in line with other authors [36,37]. Different authors have conducted comprehensive studies that examined the effects of SARS-CoV-2 on pregnancy outcomes. Rizzo et al., found that in pregnancies affected by SARS-CoV-2 infection during the second half, fetal growth and growth velocity between the second and third trimesters were similar compared to unexposed pregnancies. These data suggest that SARS-CoV-2 infection during pregnancy likely does not increase the risk of fetal growth restriction [37]. Mappa et al. provide insights into the SARS-CoV-2’s influence on fetal brain development in asymptomatic cases. Their study revealed no significant impact on cortical development or brain growth in mildly symptomatic pregnant women [38]. Only 7.6% of the infected women had mild symptoms, compared with previous proportions ranging from 20% to 26% [4]. We believe that the underlying reason for the moderate symptoms identified in the cases we have studied is probably due to the specific features of the omicron variant, which, although more contagious, tends to present less severity and associated symptomatology, in line with other authors [39].

In the comparison of the risk factors, including advanced maternal age, maternal obesity, hypertension, and gestational diabetes, no significant differences were found either between the infected and non-infected women [9,17,36,40]. Our results coincide with those of studies in the UK, Nordic countries, and Italy [41,42,43], where few complications were observed in pregnant women without associated comorbidity; these data confirm that the absolute risk of admission of pregnant women due to severe COVID-19 is low.

Our findings regarding PROM contrast with the reported literature [6,21,44,45], which often associate SARS-CoV-2 infection with a higher frequency of PROM. One potential contributing factor to this incongruity could be the link between PROM and prematurity. In our study, an interesting observation is that we had a notably lower rate of preterm births. This could be attributed to the fact that cases requiring specialized care are referred to a designated hospital. In our case, the multivariate model demonstrated a direct link between prematurity and PROM. Therefore, a higher number of preterm births could align our PROM outcomes with other studies’ findings. Consequently, the altered population composition might influence the observed relationship between SARS-CoV-2 infection and PROM, warranting a nuanced interpretation of our findings.

Consistent with other studies, we also observed an increased incidence of NICU admissions among infants born to infected mothers [7,19,21,23]. Additionally, these neonates exhibited extended NICU stays compared to those born to non-infected mothers. The relationship between maternal SARS-CoV-2 infection and increased admissions to the NICU may be attributed to factors such as vertical transmission, perinatal complications, prematurity, respiratory issues, altered immune responses, isolation protocols, maternal treatments, and maternal complications [21]. In our case, the multivariate model revealed that NICU admissions were associated with both maternal infection and prematurity. NICU admissions in neonates during the COVID-19 pandemic seem driven more by prematurity than the virus itself. Increased preterm births and NICU admissions among COVID-19-affected women have been noted [46]. In line with other authors, research suggests that heightened respiratory problems in neonates stem from preterm birth rather than neonatal COVID-19 [47].

On the contrary, the rates of maternal mortality, ICU admission, prematurity, and antepartum death have been reported to be high and significantly differ between women with and without COVID-19 [15,19]. A possible explanation for the contrasting results could be that many adverse outcomes between infected and non-infected women have generally occurred more frequently in low- and middle-income countries or in pregnant women with scarce socioeconomic resources, in which disparities are reflected in the quality of life [9] and access to healthcare [48]. Accordingly, women with a higher level of health literacy may adopt preventive measures against infection in greater proportion than those with a low level of health literacy, as previously observed [49]. Another possible explanation could be the study design. Retrospective studies based on data from the Centers for Disease Control and Prevention in the US [15,19] had 64.5% missing data during pregnancy compared with prospective studies conducted in the UK, Nordic countries, and Italy with no missing cases, consistent with our results. Furthermore, more than half of the women infected may have been infected with the omicron variant, which generally resulted in fewer hospital admissions, deaths, and complications [17].

### 4.2. Obstetric and Neonatal Outcomes between Vaccinated and Unvaccinated Women

To date, there are limited studies comparing the outcomes of pregnant women who received COVID-19 vaccines with those of other infected, unvaccinated pregnant women [30,50,51]. The study witnessed a vaccination rate of 30%. It is noteworthy that our case selection extends from the inception of the pandemic until a year preceding its conclusion. Notably, approved vaccinations for pregnant individuals became accessible in Spain as of July 2021. Consequently, within the timeframe ranging from March 2020 to that date, a subset of the population remained unvaccinated due to this criterion. Our deliberate inclusion of cases from the pandemic’s initiation is rooted in our intention to meticulously compare outcomes between vaccinated and non-vaccinated individuals. Given that the first vaccinated individuals gave birth from November 2021 onwards, the study cohort’s size contracts to encompass 453 women, with a vaccination rate of 71% (322/453).

In our study, the rates of adverse outcomes after delivery (pregnancy outcomes) among the women who received at least one dose of COVID-19 vaccine during pregnancy were similar to those among the women who were not vaccinated, with no significant differences. Our results coincide with previous data [27]. Despite having a considerably larger sample size, previous studies did not also show any differences.

In recent studies, there was no increased risk of preeclampsia observed in vaccinated pregnant women [31,52] compared with that in unvaccinated women, in contrast with our results [6]. This discrepancy highlights the need for further research to investigate any potential association between COVID-19 vaccination and the development of preeclampsia [53]. Factors such as variations in study populations, methodologies, and timing of vaccination may contribute to the contrasting results. Understanding the underlying mechanisms and potential risk factors associated with preeclampsia in vaccinated pregnant women is crucial for optimizing antenatal care and guiding future vaccination strategies.

Contrary to other reports [54], we did find a significant difference in the proportions of preterm births and PROM between vaccinated and unvaccinated women. The cause of the observed differences between vaccinated and unvaccinated women may be multifactorial. Possible explanations may include differences in sample characteristics, interactions between vaccination and other risk factors, biological effects of vaccination, and contextual factors. However, it is important to note that investigating these potential causes fell outside the scope of our study objectives. In line with the same argument, we also examined variables such as postpartum hemorrhage and increased maternal oxygen supplementation after birth. While these findings suggest potential associations between vaccination and these factors, it is important to note that exploring the multifactorial explanation for these relationships fell outside the scope of our study objectives.

The safety of the vaccine in pregnant and lactating populations has been previously evaluated. Although the data provided are still limited to mRNA vaccines, there appears to be no evidence of increased rates of miscarriages, fetal malformations, or fertility problems [6,10]. Herein, we did not observe any malformations in the vaccinated group. Evidence related to the benefits of vaccination against COVID-19 during pregnancy continues to accumulate, with beneficial results for both mothers and their children [6,9,26]. The findings reinforce the importance of communicating the risks of COVID-19 during pregnancy, benefits of vaccination, and safety and effectiveness of COVID-19 vaccination during pregnancy [6].

In our study, the incidence of SARS-CoV-2 infection almost doubled from the first to the second year of follow-up, consistent with published reports and despite the fact that the vaccination campaign had already been implemented by the end of June 2021 [17]. The spread of the omicron variant could explain this increase. This fact confirms that obstetric outcomes are not necessarily similar in all waves, as observed in the first wave with the alpha variant [6]. At the time of writing, we do not have solid data on how the omicron variant has affected pregnant women, although our results agree with previous reports that the effects are less severe with this variant than with the alpha variant [17].

The study has some limitations. Owing to the retrospective observational study design, it was possible to randomize the non-infected cohort, which implies that the possibility of residual confounding could not be ruled out. Consequently, the study could not establish causality but only association between the factors analyzed, and it was not possible to exclude the presence of unknown or unobserved factors that could explain the results and extend beyond our analysis scope. In addition, the sample size was relatively modest, which means that the statistical power is somewhat limited, especially when identifying rare events. Nevertheless, we included all positive cases managed in our hospital during the study period without any loss to follow-up and exceeding the estimated sample size for sufficient representation and pre-determined power of the study. Regarding data collection from medical records, the possible inaccuracy in the quality of data reporting was mitigated by allowing only a single reviewer with sufficient clinical experience to review and classify the data in accordance with the initial protocol. Finally, our hospital attends preterm deliveries from 34 weeks of gestation due to its intermediate care neonatal unit, so the low prevalence of preterm deliveries may be explained by this and may underestimate the outcome. Finally, the study does not encompass data regarding pregnant mothers with prior COVID-19 infections and their potential antibody development, which represents a significant gap in our research. This omission hinders our capacity to comprehensively examine the impact of previous infections on antibody responses during pregnancy, an area of investigation that could yield valuable insights into the broader ramifications of prior infection on immune dynamics in this unique scenario. Importantly, it is worth noting that at the time our study was conducted, robust evidence indicating the likelihood, commonness, or significance of reinfection was not prevalent, as corroborated by the findings of Stein et al. [55]. Furthermore, we were unable to compare NICU admissions for SARS-CoV-2 infections with those caused by common viral infections like influenza due to the unavailability of relevant data, limiting our ability to provide a comprehensive comparative analysis of neonatal outcomes across different viral infections.

## 5. Conclusions

In our university hospital, obstetric and perinatal complications in the women infected with SARS-CoV-2 after delivery were rare, and the symptomatology was mild. The severe adverse outcomes did not significantly differ between the infected and non-infected groups. However, SARS-CoV-2 infection can still pose significant risks to pregnant women and their offspring. While we did not observe any vertical transmission of the infection during hospital admission, we did identify a heightened NICU admission rate among neonates born to infected mothers, predominantly preterm infants. Notably, the primary reason for NICU admission was not respiratory complications stemming from maternal infection. Our study has revealed a higher incidence of preterm births, premature rupture of membranes, preeclampsia, and postpartum hemorrhage in vaccinated women compared to those who were not vaccinated. These findings suggest a potential association between vaccination and these adverse outcomes. Further research is needed to better understand the underlying factors contributing to these associations and to evaluate the overall risk–benefit profile of vaccination in pregnant women. Nevertheless, given the clear preventive value of vaccines, many pregnant women and their offspring continue to lack access to this public healthcare, increasing the risk of morbidity and mortality from COVID-19. Accordingly, we advocate for greater determination in the field of vaccine diplomacy to mitigate preventive and care inequalities and in the field of research to continue to deepen knowledge aimed at the prevention and control of COVID-19 worldwide. This study contributes to the knowledge base for informed decision-making in managing COVID-19 during pregnancy.

## Figures and Tables

**Figure 1 healthcare-11-02833-f001:**
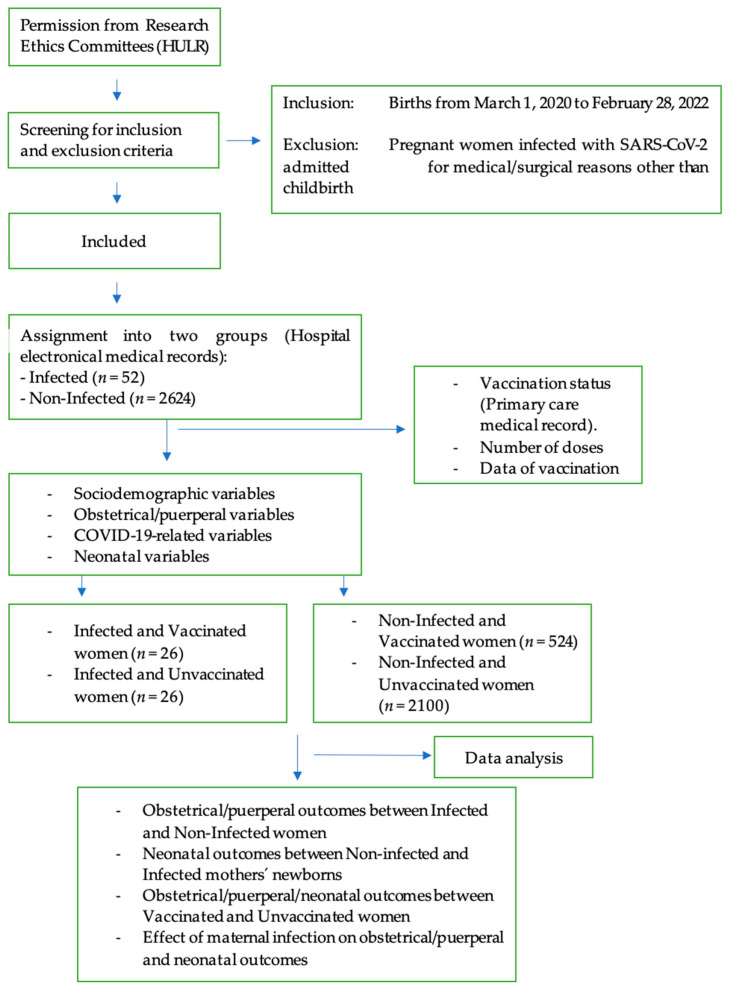
Data collection flowchart.

**Table 1 healthcare-11-02833-t001:** Comparison of the vaccination status and sociodemographic characteristics between the women non-infected and infected to SARS-CoV-2 (N = 2676).

		SARS-CoV-2 Infection				
	Total	No	Yes	OR ^1^	95% CI ^1^	*p*-Value *^,^^‡^
	N = 2676	*n* = 2624 (98.1%)	*n* = 52 (1.9%)			
Maternal age				1.00	1.0, 1.0	0.911
Mean (SD)	31.19 (6.12)	31.19 (6.11)	31.29 (6.58)			
Median (IQR)	32.00 (27.00, 36.00)	32.00 (27.00, 36.00)	31.50 (27.00, 35.25)			
Range	14.00, 50.00	14.00, 50.00	18.00, 46.00			
Parity						
Primiparous	1484 (55.5%)	1463 (55.8%)	21 (40.4%)	1.00	—	
Multiparous	1192 (44.5%)	1161 (44.2%)	31 (59.6%)	1.91	1.1, 3.3	0.03
Country of origin						
Spain	1922 (72%)	1886 (72%)	36 (69%)	1.00	—	
Foreign	754 (28%)	738 (28%)	16 (31%)	1.13	0.6, 2.0	0.675
Vaccination status						
No	2126 (79.5%)	2100 (80%)	26 (50%)	1.00	—	
Yes	550 (20.5%)	524 (20%)	26 (50%)	4.04	2.3, 6.9	<0.001
Dose						
Unvaccinated	2126 (79%)	2100 (80%)	26 (50%)	1.00	—	
Incomplete regimen	108 (4.0%)	101 (3.8%)	7 (13%)	5.63	2.2, 13.0	<0.001
Complete regimen	442 (17%)	423 (16%)	19 (37%)	3.64	2.0, 6.6	<0.001

^1^ OR = odds ratio, CI = confidence interval; * Chi-squared test; ^‡^ Student *t*-test.

**Table 2 healthcare-11-02833-t002:** Comparison of the obstetric characteristics between the non-infected and infected women with SARS-CoV-2 (N = 2676).

		SARS-CoV-2 Infection				
	Total	No	Yes	OR ^1^	95% CI ^1^	*p*-Value *^,‡^
	N = 2676 (100%)	*n* = 2624 (98.1%)	*n* = 52 (1.9%)			
Preterm birth < 37 weeks						
No	2466 (92)	2419 (92.2)	47 (90.4)	1.00	—	
Yes	210 (7.8)	205 (7.8%)	5 (9.6%)	1.29	0.4, 2.9	0.633
Premature rupture of membranes						
No	1564 (58)	1523 (58)	41 (79)	1.00	—	
Yes	1112 (42)	1101 (42)	11 (21)	0.41	0.2, 0.7	0.004
Antepartum hemorrhage						
No	2651 (99)	2600 (99.1)	51 (98.1)	1.0	—	
Yes	25 (0.9)	24 (0.9)	1 (1.9)	2.12	0.1, 10	0.465
Postpartum hemorrhage						
No	2655 (99.2)	2604 (99.2)	51 (98.1)	1.00	—	
Yes	21 (0.8)	20 (0.8)	1 (1.9)	2.64	0.1, 13	0.365
Caesarean section						
No	2149 (79.3)	2105 (80.2)	44 (84.6)	1.00	—	
Yes	527 (19.7)	519 (19.8)	8 (15.4)	0.72	0.3, 1.5	0.432
Instrumental delivery						
No	2358 (88)	2316 (88)	42 (81)	1.00	—	
Yes	318 (12)	308 (12)	10 (19)	1.81	0.8, 3.5	0.103
Fetal distress						
No	2642 (99)	2592 (98.8)	50 (96.2)	1.00	—	
Yes	34 (1.3)	32 (1.2)	2 (3.8)	3.23	0.5, 11	0.113
Preeclampsia/eclampsia/gestational hypertension/HELLP						
No	2595 (97)	2544 (97)	51 (98.1)	1.00	—	
Yes	81 (3.0)	80 (3.0)	1 (1.9)	0.61	0.0, 2.9	0.642
Gestational diabetes						
No	2479 (93)	2431 (92.6)	48 (92.3)	1.00	—	
Yes	197 (7.4)	193 (7.4)	4 (7.7)	1.00	0.3, 2.6	0.927
Presence of SARS-CoV-2 signs/symptoms						
No	2672 (99.9)	2624 (100)	48 (92.3)			
Yes	4 (0.1)	0 (0)	4 (7.7)			
ICU admission						
No	2661 (99.4)	2610 (99.5)	51 (98.1)	1.00	—	
Yes	15 (0.6)	14 (0.5)	1 (1.9)	3.72	0.2, 19	0.215
Days in ICU						
Mean (SD)	1.33 (0.90)	1.14 (0.53)	4.00 (NA)			
Median (IQR)	1.00 (1.00, 1.50)	1.00 (1.00, 1.00)	4.00 (4.00, 4.00)			
Range	0.00, 4.00	0.00, 2.00	4.00, 4.00			
Supplemental oxygen						
No	2669 (100%)	2618 (99.8%)	51 (98.1%)	1.00	—	
Yes	7 (0.3%)	6 (0.2%)	1 (1.9%)	8.61	0.4, 51	0.049
Asthma						
No	2644 (98.8)	2594 (98.9)	50 (96,2)	1.00	—	
Yes	32 (1.2)	30 (1.1)	2 (3.8)	1.19	0.8, 1.7	0.075
BMI > 30 kg/m^2^						
No	2418 (90.4)	2370 (90.3)	47 (90.4)	1.00	—	
Yes	259 (9.6)	254 (9.7)	5 (9.6)	1.21	0.8, 1.3	0.987
Hyperthyroidism						
No	2632 (98.4)	2581 (98.4)	51 (98.1)	1.00	—	
Yes	44 (1.6)	43 (1.6)	1 (1.9)	0.17	0.3, 0.6	0.875
Hypothyroidism						
No	2249 (84.0)	2205 (84.0)	44 (84.6)	1.00	—	
Yes	427 (26.0)	419 (26.0)	8 (5.4)	0.10	0, 0.3	0.909
Smoker						
No	2479 (92.6)	2431 (92.6)	48 (92.3)	1.00	—	
Yes	197 (7.4)	193 (7.4)	4 (7.7)	1.09	0.7, 1.3	0.876

^1^ OR = odds ratio, CI = confidence interval; * Chi-squared test; ^‡^ Student *t*-test; PROM: premature rupture of membranes; ICU: intensive care unit; BMI: body mass index.

**Table 3 healthcare-11-02833-t003:** Comparison of neonatal characteristics between non-infected and infected mothers’ newborns to SARS-CoV-2 (N = 2676).

		SARS-CoV-2 Infection				
	Total	No	Yes	OR ^1^	95% CI ^1^	*p*-Value *^,‡^
	N = 2676 (100%)	*n* = 2624 (98.1%)	*n* = 52 (1.9%)			
Small for gestational age						
No	2600 (97.2%)	2550 (97.2%)	50 (96.2%)	1.00	—	
Yes	76 (2.8%)	74 (2.8%)	2 (3.8%)	1.39	0.2, 4.6	0.661
Large for gestational age						
No	2354 (88%)	2306 (88%)	48 (92.3%)	1.00	—	
Yes	322 (12%)	318 (12%)	4 (7.7%)	0.61	0.2, 1.5	0.336
Respiratory distress						
No	2637 (98.5%)	2587 (98.6%)	50 (96.4%)	1.00	—	
Yes	39 (1.5%)	37 (1.4%)	2 (3.8%)	2.83	0.4, 9.5	0.165
NICU admission						
No	2465 (92.1%)	2422 (92.3%)	43 (83%)	1.00	—	
Yes	211 (7.9%)	202 (7.7%)	9 (17%)	2.47	1.1, 5.0	0.014
Length of stay >4 days						
No	2647 (98.9%)	2598 (99.0%)	49 (94%)	1.00	—	
Yes	29 (1.1%)	26 (1.0%)	3 (5.8%)	6.07	1.4, 18	0.004
Ventilator support						
No	2639 (98.6%)	2589 (98.7%)	50 (96%)	1.00	—	
Yes	37 (1.4%)	35 (1.3%)	2 (3.8%)	3.00	0.5, 10	0.143
Neonatal death						
No	2670 (99.8%)	2618 (99.8%)	52 (100%)	1.00	—	
Yes	6 (0.2%)	6 (0.2%)	0 (0%)	0.00		0.984

^1^ OR = odds ratio, CI = confidence interval; * Chi-squared test; ^‡^ Student *t*-test; NICU: neonatal intensive care unit.

**Table 4 healthcare-11-02833-t004:** Multivariate logistic regression models for preterm birth (<37 w), PROM, C-section, ICU admission, and NICU admission (N = 2676).

Preterm Birth < 37 Weeks				PROM			C-Section			ICU Admission			NICU Admission		
	OR ^1^	95% CI ^1^	*p*-Value	OR ^1^	95% CI ^1^	*p*-Value	OR ^1^	95% CI ^1^	*p*-Value	OR ^1^	95% CI ^1^	*p*-Value	OR ^1^	95% CI ^1^	*p*-Value
SARS-CoV-2 infection															
No	—	—		—	—		—	—		—	—		—	—	
Yes	1.21	0.41, 2.78	0.700	0.39	0.17, 0.67	0.003	0.77	0.34, 1.66	0.6	2.81	0.13, 17.90	0.4	2.87	1.23, 6.06	0.009
Vaccination status															
No	—	—		—	—		—	—		—	—		—	—	
Yes	1.42	0.97, 1.88	0.065	1.22	1.00, 1.47	0.048	0.89	0.74, 1.20	0.6	1.38	0.41, 4.22	0.6	0.91	0.60, 1.26	0.5
Preterm birth < 37 weeks															
No	—	—	—	—	—		—	—		—	—		—	—	
Yes	—	—	—	1.43	1.07, 1.89	0.015	1.32	0.90, 1.78	0.2	5.37	1.62, 15.50	0.003	6.73	4.73, 9.41	<0.001

^1^ OR = odds ratio, CI = confidence interval; PROM: premature preterm rupture of membranes; NICU: neonatal intensive care unit; adjusted model by maternal age, parity, and country of origin.

## Data Availability

The authors have checked to make sure that our submission conforms as applicable to the Journal’s statistical guidelines. The statistics were checked prior to submission by two expert statisticians, Carlos Vergara Hernández (estadistica_fisabio@gva.es) and Maria Jose Caballero Mateos (caballero_mjomat@gva.es). The authors affirm that the methods used in the data analyses are suitably applied to their data within their study design and context, and the statistical findings have been implemented and interpreted correctly. Furthermore, the authors agree to take responsibility for ensuring that the choice of statistical approach is appropriate and is conducted and interpreted correctly as a condition to submit to the Journal. A bivariate descriptive analysis was performed, comparing women with and without confirmed SARS-CoV-2 infection during pregnancy using the Chi-square test. A logistic regression analysis was complementarily conducted to determine whether SARS-CoV-2 infection increases the risk of adverse obstetric and perinatal outcomes.

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
