# Peer review of "Perinatal Outcomes at Birth in Women Infected and Non-Infected with SARS-CoV-2: A Retrospective Study"

_healthcare, 2023, doi:10.3390/healthcare11212833_

Round 1

Reviewer 1 Report (Previous Reviewer 2)

Comments and Suggestions for Authors

In spite of the revisions provided by the authors, I do not think that this manuscript is novel considering the previous landscape of publications in this subject matter of which there are probably 100s. Adding another location of which covid-19 impacted pregnant persons unfortunately does not add much further to our understanding of pregnancy outcomes.

Author Response

Reviewer 2 Report (New Reviewer)

Comments and Suggestions for Authors

I read it with great interest, but I have raised several concerns.

#1. Coronavirus disease (COVID-19) -> Coronavirus disease 2019 (COVID-19)

#2. SARS-CoV-2 -> Please use full name with abbreviation.

#3. Please add the hypothesis of this study in introduction.

#4. (maternal obesity, asthma, smoker), -> (maternal obesity, asthma, or smoker),

The authors have to receive English editing service.

#5. The computation of odds ratios and their cor-responding confidence intervals, accompanied by p-values derived from log-odds through the utilization of the Wald test, was executed by employing a series of logistic  regression models featuring a sole independent variable.  -> The authors have to cite the statistical guideline(DOI: https://doi.org/10.54724/lc.2022.e3).

#6. In Table 1, it is customary to present the Odds Ratio rounded to two decimal places.

#7. I want to see stratified analysis from vaccination type (mRNA, adenoviral...).

#8. Please limitations in more detail.

Comments on the Quality of English Language

I read it with great interest, but I have raised several concerns.

#1. Coronavirus disease (COVID-19) -> Coronavirus disease 2019 (COVID-19)

#2. SARS-CoV-2 -> Please use full name with abbreviation.

#3. Please add the hypothesis of this study in introduction.

#4. (maternal obesity, asthma, smoker), -> (maternal obesity, asthma, or smoker),

The authors have to receive English editing service.

#5. The computation of odds ratios and their cor-responding confidence intervals, accompanied by p-values derived from log-odds through the utilization of the Wald test, was executed by employing a series of logistic  regression models featuring a sole independent variable.  -> The authors have to cite the statistical guideline(DOI: https://doi.org/10.54724/lc.2022.e3).

#6. In Table 1, it is customary to present the Odds Ratio rounded to two decimal places.

#7. I want to see stratified analysis from vaccination type (mRNA, adenoviral...).

#8. Please limitations in more detail.

Author Response

"Please see attachment"

Reviewer 3 Report (New Reviewer)

Comments and Suggestions for Authors

The authors report on Perinatal Outcomes at Birth in Women Infected and Non-Infected with SARS-CoV-2.

I appreciated that they arrived to interesting and useful conclusions.

So, the authors found that the prematurity was not affected by SARS-CoV-2.

Neither cesarian section was not increased in frequency in infected mothers.

Other complications as: abruptio placentae, antepartum/postpartum hemorrhage, fetal distress, fetal growth restriction, preeclampsia, gestational diabetes, or maternal death, occurred in similar rats in non-infected and non-infected patients.

It was also interesting the fact that the vertical viral transmission was very low.

The viral presence in vaginal secretion and milk, were, again, very low.

So, I concluded it is an interesting and useful study!

Good work!

I have only one comment:

The authors found that NICU admission / period was significantly longer in infected patients, fact expected. I suggest to complete the study with a comparison between the impact of SARS-CoV-2 infection and a banal viral infection as influenza, on this parameter, to more clearly reflect the real impact of the coronavirus on the neonates.

Round 2

Reviewer 2 Report (New Reviewer)

Comments and Suggestions for Authors

This is an excellent paper.

Author Response

Thank you for your feedback on our manuscript.

All your comments have been taken into consideration and amendments have been done on the previous version of the manuscript.

This manuscript is a resubmission of an earlier submission. The following is a list of the peer review reports and author responses from that submission.

Round 1

Reviewer 1 Report

Comments and Suggestions for Authors

The manuscript authored by Rafael Vila-candel. et.al present a study that conducted at university hospital in Spain to investigate the correlation between SARS-CoV-2 infection during childbirth and negative perinatal outcomes among women who were either vaccinated or unvaccinated. The abstract is well-written and effectively conveys the key findings and implications of the study. However, the manuscript requires additional revision to adequately address the issues raised in the review. Further revision of interpretation of results, clarity and accuracy of data could greatly enhance the manuscript impact and make it more engaging for readers. 

·       Line 50-51: "In addition, the virus has not been detected in vaginal secretions or breast milk."- Multiple studies have reported the detection of SARS-CoV-2 RNA in breast milk. Please review the literature to ensure the accuracy of this statement.

·       To enhance the clarity of the result and assist the audience in comprehending the information more easily, please incorporate graphs to represent statistically significant data. In addition, it would be beneficial to mark the significant values using asterisks, as this will provide a visual cue and make it simpler for readers to identify important points.

·       Line 91: "Different clinical trials evaluating the efficacy of mRNA vaccination have consistently excluded pregnant women. However, it would be valuable to explore the potential benefits and safety profiles of other vaccine platforms, such as adeno virus-based or recombinant vaccines, in this population.

·       It is recommended to present the data collection section in the form of a table to enhance readability for the audience. A table can effectively organize the information and make it easier to understand. Please provide a table that includes the relevant details of the data collection process.

·       The authors are requested to provide additional information regarding the inclusion of pregnant mothers with previous COVID-19 infections and their likelihood of developing antibodies in the study. This data is important to better understand the impact of prior infection on antibody development during pregnancy.

·       To ensure clarity for readers, please provide the full form of PROM, and then follow it with the abbreviated form in the text.

Reviewer 2 Report

Comments and Suggestions for Authors

Thanks for the opportunity to review this paper on the differences between outcomes of those with covid 19 vaccination status in pregnant persons. 

General comments are that these findings are not novel as it has been reported in several papers to date. The low vaccination rate of about 30% or less in the population of Spain has the authors falsely attributing outcomes that were not severe to the covid 19 vaccination status. There are low numbers compared to other populations worldwide and also because of the type of hospital where the patients were seen, there is a good chance that those were poorer outcomes were referred outside of the institution of interest. This therefore skews the findings a great deal. 

I would suggest that the authors refer to some past references to put this paper into better context especially since several of the protocols from 2020-2022 changed significantly including removal of mask mandates, loosening of lockdowns, etc. 

Futterman I, Toaff M, Navi L, Clare CA. COVID-19 and HELLP: Overlapping Clinical Pictures in Two Gravid Patients. AJP Rep. 2020 Apr;10(2):e179-e182. doi: 10.1055/s-0040-1712978. Epub 2020 Jun 18. PMID: 32566368; PMCID: PMC7302930.

Liu, C., Andrusier, M., Silver, M., Applewhite, L., & Clare, C. A. (2021). Effect of SARS-CoV-2 Infection on Pregnancy Outcomes in an Inner-City Black Patient Population. Journal of community health46(5), 1029–1035. https://doi.org/10.1007/s10900-021-00988-z

Reviewer 3 Report

Comments and Suggestions for Authors

The authors investigated the perinatal outcomes in infected and non-infected SARS-CoV-2 pregnant women. 

1.  What does this paper add up to the existing knowledge on SARS-CoV-2 infection in pregnancy? 

2.  The sample size (COVID-19 infected mothers) was rather small, and majority of these women were asymptomatic. Suggest to increase the sample size and to include women with various COVID-19 severity.

3.  Having mere SARS-CoV-2 RT-PCR positive does not mean transplacental/ vertical transmission. Hence expectedly, no different in perinatal outcome when compared with healthy normal population.

4.  What about other maternal comorbidities such as maternal obesity, GBS carrier, systemic diseases e.g. SLE, thyroid disease, or anaemia in pregnancy etc.? These could serve as potential confounding factors that could affect pregnancy outcomes.